# Neurally-Adjusted Ventilatory Assist (NAVA) versus Pneumatically Synchronized Ventilation Modes in Children Admitted to PICU

**DOI:** 10.3390/jcm10153393

**Published:** 2021-07-30

**Authors:** Pravin Sugunan, Osama Hosheh, Mireia Garcia Cusco, Reinout Mildner

**Affiliations:** 1Paediatric Intensive Care Unit, Birmingham Children’s Hospital, Birmingham B4 6NH, UK; sdnh82@yahoo.com (P.S.); osama.hosheh@nhs.net (O.H.); 2Paediatric Intensive Care Unit, Royal Bristol Children’s Hospital, Bristol BS2 8BJ, UK; mireia.garciacusco@gmail.com

**Keywords:** neurally adjusted ventilatory assist, NAVA, invasive mechanical ventilation, weaning, children, paediatric intensive care

## Abstract

Traditionally, invasively ventilated children in the paediatric intensive care unit (PICU) are weaned using pneumatically-triggered ventilation modes with a fixed level of assist. The best weaning mode is currently not known. Neurally adjusted ventilatory assist (NAVA), a newer weaning mode, uses the electrical activity of the diaphragm (Edi) to synchronise ventilator support proportionally to the patient’s respiratory drive. We aimed to perform a systematic literature review to assess the effect of NAVA on clinical outcomes in invasively ventilated children with non-neonatal lung disease. Three studies (*n* = 285) were included for analysis. One randomised controlled trial (RCT) of all comers showed a significant reduction in PICU length of stay and sedative use. A cohort study of acute respiratory distress syndrome (ARDS) patients (*n* = 30) showed a significantly shorter duration of ventilation and improved sedation with the use of NAVA. A cohort study of children recovering from cardiac surgery (*n* = 75) showed significantly higher extubation success, shorter duration of ventilation and PICU length of stay, and a reduction in sedative use. Our systematic review presents weak evidence that NAVA may shorten the duration of ventilation and PICU length of stay, and reduce the requirement of sedatives. However, further RCTs are required to more fully assess the effect of NAVA on clinical outcomes and treatment costs in ventilated children.

## 1. Introduction

Around the world, the annual admission rates of children to paediatric intensive care units (PICU) range between 76 and 150 per 100,000 people [1,2,3], with between 42% and 63% receiving invasive mechanical ventilation. In the United Kingdom (UK) and Ireland, around 20,000 children are admitted annually to a PICU [1]. Of these, 62% receive invasive mechanical ventilation with 44,640 intubated days (average 3.6 intubated days per patient). Based on the reference cost for a PICU bed-day of GBP 2178 [4], the estimated total annual cost of PICU treatment for invasively ventilated children is GBP 97 million.

During recovery in the PICU, invasively ventilated children are traditionally weaned (liberated) from mechanical ventilation using pneumatically-triggered ventilation modes to achieve patient-ventilator synchrony (P-VS) with a pre-set fixed level of support [5]. The level of positive end-expiratory pressure (PEEP) is usually set according to the amount of additional inspired oxygen required by the patient, the patient’s haemodynamic condition, and the presence of airway malacia. Evidence is lacking regarding the best method of weaning [5].

Patient-ventilator asynchrony (P-VA) and discomfort occur in up to 30% of children [6,7,8,9,10,11,12]. In critically ill adults, P-VA prolongs weaning time, with a significantly increased duration of ventilation, length-of-stay (LOS), and risk of complications [13,14,15,16]. Positive pressure ventilation is also known to induce diaphragm atrophy [17]. Furthermore, in children, weaning may be adversely affected by over-assistance and over-sedation [8].

Neurally adjusted ventilatory assist (NAVA) is a relatively new weaning mode, available only with Maquet/Getinge Group-brand ICU ventilators [18]. During NAVA, the electrical activity of the diaphragm (Edi) is measured using a special nasogastric tube (Edi catheter), and the signal is used to achieve P-VA. The level of support provided is in proportion to the Edi (proportional assist). The amount of PEEP that patients receive can be adjusted according to the Edi level during expiration [19]. Physiological studies in adults suggest that NAVA may preserve diaphragm function better than the conventional pressure support (PS) mode of ventilation [20]. The risk of over-assistance may be lower due to the intact neural control mechanisms [21,22]. The use of NAVA can be hampered by difficulties in obtaining the Edi signal and unfamiliarity with NAVA amongst PICU healthcare professionals, requiring additional training [22,23]. Although it is CE-marked (compliant with European Union legislation and safety requirements), it has not been widely adopted so far in PICUs to facilitate weaning.

A systematic review published in 2016 [24] showed that most of the relevant published paediatric studies compared ventilatory physiological parameters between NAVA and conventional ventilation modes. The use of NAVA has been associated with reduced PV-A, peak inspiratory pressure (PIP), and fractions of inspired oxygen (FiO_2_), as well as improvements in patient comfort, work of breathing, and gas exchange. Relatively few studies, however, have focused on the effects of NAVA on clinical outcomes.

Our aim was to review the available evidence on the effects of weaning from invasive mechanical ventilation in NAVA compared to conventional ventilation modes on clinical outcomes in children admitted to paediatric intensive care units.

## 2. Materials and Methods

In carrying out the study, we followed the Preferred Reporting Items for Systematic reviews and Meta-Analyses (PRISMA) statement [25]. The following sources were searched: MEDLINE (1947–August 2020), EMBASE (1947–August 2020), CINAHL (1937–August 2020), the Cochrane Library, and the Turning Research Into Practice (TRIP) database. For publications in languages other than English, Dutch, French, or Spanish, translation was planned.

We searched for randomised controlled trials (RCTs) and cohort studies which included invasively ventilated children (aged from 4 weeks post-term to 18 years) admitted to a PICU with non-neonatal lung disease, comparing weaning from invasive mechanical ventilation in NAVA versus other pneumatically-controlled conventional modes of ventilation.

Studies were included that reported one or more of the following clinical outcomes: duration of ventilation, ventilator-free days, PICU length of stay, sedation use, re-intubation within 24 h, PICU mortality, complications of ventilation, and cost, expressed as mean or median difference, or risk difference for reintubation and mortality. Details of the full search strategy are included in Table A1.

Three authors (PS, RM, and OH) independently reviewed the identified records, using the title and abstract. The full texts of studies meeting inclusion criteria were then reviewed in detail. The following data were extracted using a standardised proforma: study design, patient population, age, setting, type of intervention and control, and outcomes. After independent assessments for eligibility, the authors agreed on studies for inclusion.

We planned to provide a structured narrative review of included studies describing study design, setting, participants, interventions, comparators (NAVA versus conventional weaning modes), and main outcomes. We did not plan to carry out a formal data synthesis between studies, as we expected to find only a small number of studies for analysis. For randomised controlled trials, the Cochrane risk-of-bias [26] methodology was used to assess for bias in the included studies.

## 3. Results

A total of 478 articles were identified by our search, as summarised in the flow chart in Figure 1. After the removal of duplicates, 459 records were assessed for eligibility. Four studies were included in the review.

One was a randomised controlled trial [26] and three were cohort studies [27,28,29,30]. After correspondence with the authors, it was confirmed that the patients in the study by Piastra et al., published in abstract form in 2011 [28], were included in the study by the same authors published in 2014 [29]. Therefore, three were suitable for final inclusion and analysis, with a total population of 275 patients (Table 1).

Kallio et al. [27] reported a randomised controlled trial of 170 children, from newborn to 16 years of age, expected to be ventilated in PICU for at least 30 min. In the intervention group, patients were weaned using NAVA versus patient-triggered time-cycled, synchronised intermittent mandatory ventilation with pressure-control or synchronised intermittent mandatory ventilation with pressure-regulated volume control (SIMV-PC or SIMV-PRVC) in the control group. The sedation target (sedation agitation score level 4) was the same in both groups. In this study, 77% of patients were ventilated for post-operative care. The median duration of ventilation was 3.3 h in the NAVA group versus 6.6 h in the control group (not significantly different). The median PICU length of stay in the NAVA group was 49.5 h versus 72.8 h in controls. This was significant (*p* = 0.03) in the per-protocol analysis. Sedative use in units per hour in the group of patients admitted for reasons other than post-operative care was on average 1.43 units per hour lower (95% confidence interval (CI) −2.79 to −0.07; *p* = 0.03) in the NAVA group. There was no difference in treatment complications, including deaths. The risk of bias for the study by Kallio et al. [27] is reported in Table 2.

Piastra et al. [29] conducted a retrospective cohort study of 30 infants of less than 1 year of age with acute respiratory distress syndrome (ARDS) determined according to American/European Consensus criteria, requiring rescue high-frequency oscillatory ventilation (HFOV). Patients were matched in a 1:2 ratio by age, weight, and PaO_2_/FiO_2_ ratio, with 10 patients weaned using NAVA and 20 patients using pressure support ventilation (PSV) according to a standard local protocol that remained unchanged during the study period. The same doses of sedatives were given in both cohorts. The mean duration of weaning was 41 (±17) h in the NAVA cohort versus 72.5 (±44) h in the PSV cohort (*p* = 0.011). The was no significant difference in PICU length of stay. The COMFORT sedation score was 18.1 (±2.1) in the NAVA cohort and 25.3 (±7) in the PSV cohort (*p* = 0.004). No deaths occurred during the study period. Four patients (two in each cohort) died at a later stage. The authors reported no technical problems with the NAVA equipment.

Sood et al. [30] reported a cohort study of 75 children undergoing heart surgery on cardio-pulmonary bypass requiring at least 96 h of full post-operative ventilation (SIMV-PRVC + PSV). Following this period, one cohort was weaned using NAVA (*n* = 35) and the retrospective cohort was weaned using SIMV-PRVC + PSV (*n* = 40). There were no significant differences in clinical characteristics, surgical risk, or days of ventilation prior to weaning between cohorts. The primary outcome of extubation success, defined as remaining extubated for over 24 h, was 97.1% in the NAVA cohort and 80% in the SIMV-PRVC + PSV cohort (*p* = 0.032). The odds ratio for successful extubation in the NAVA cohort compared to the SIMV-PRVC+PSV cohort was 8.5 (95% CI 1.01 to 71.8). Median total duration of ventilation was 9 days (Interquartile Range [IQR] 4 days) in the NAVA cohort versus 11 days (IQR 7.5 days, *p* = 0.0128) in the SIMV-PRVC+PSV cohort. The median PICU length of stay was 9 days (IQR 4 days) versus 13.5 days (IQR 7.5 days, *p* < 0.0001) in the NAVA and SIMV-PRVC + PSV cohorts, respectively. The median days on fentanyl and midazolam were significantly lower in the NAVA cohort versus the SIMV-PRVC+PSV cohort (9 days (IQR: 5 days) versus 12.5 days (IQR: 7 days, *p* < 0.0001)) and 8 days (IQR: 4 days) versus 12 days (IQR: 7 days, *p* < 0.0001), respectively).

Mortality was not reported for this study. None of the included studies reported on the cost of weaning with NAVA versus conventional ventilation.

## 4. Discussion

This systematic literature review of weaning using NAVA in invasively ventilated children identified one randomised controlled trial, including 170 patients, and two cohort studies, including 105 patients, reporting the effects on pre-specified clinical outcomes. Together they provide weak evidence that weaning using NAVA may shorten the duration of ventilation and the PICU length of stay, reduce the use of sedation, and improve the rate of extubation success. No difference was found in the incidence of ventilation-associated complications or mortality. No studies reported effects on the cost of care. Although the three included studies [27,29,30] showed a shorter duration of ventilation using NAVA, in only two studies was this significant (by 24 to 48 h on average) [29,30]. In terms of the PICU length of stay, in two studies [27,30] this was reduced, by between 23 h and 4.5 days on average. In the one study that reported on the extubation success rate, this was significantly higher with NAVA. Two studies reported significant reductions in the use of sedative drugs, whereas the other study reported significantly improved levels of sedation with the same dose of sedative drugs.

A systematic review, published in 2019, which compared weaning using NAVA with PSV [31] in invasively ventilated adults, found no evidence of a reduction in the duration of ventilation or the ICU stay, with two studies included. A more recently published RCT by Kacmarek et al. [32] in invasively ventilated adults with acute respiratory failure, comparing weaning using NAVA to conventional weaning, showed a significant reduction in the duration of ventilation by 4 days on average. Liu et al. [33] recently conducted a RCT in difficult-to-wean adult ICU patients who were able to tolerate PSV and showed a significant reduction in the duration of ventilation by 5.5 days on average. The contradictory outcomes reported in the abovementioned studies could possibly be explained by differences in the timing of the introduction of NAVA and/or the severity of disease. The use of NAVA may be beneficial when commenced earlier in the weaning process and in patients with more severe respiratory failure. However, in the study by Kallio et al. [27] a benefit in terms of shortened PICU length of stay was observed in a wide range of paediatric patients. This requires further study.

A systematic review of NAVA use in neonates by Rosser at al [34] included only one study that reported clinical outcomes. Since the introduction of NAVA, many studies have been published on its physiological effects [24]. Relatively few studies have reported its effect on clinical outcomes across neonatal, paediatric, and adult intensive care patients and further RCTs are required.

One of the cohort studies included children recovering from cardiac surgery [30]. There may be physiologic effects, such as the lower peak airway pressures observed with NAVA, which can explain why these children may benefit. It is plausible that recovery may be hastened due to improved cardio-pulmonary interactions with NAVA use [35]. This area requires further study.

The significant reduction in sedative use observed with NAVA in the three studies may affect other outcomes, such as delirium, sedation withdrawal, and overall long-term health-related outcomes.

In terms of the technical difficulties encountered with the use of NAVA, Piastra et al. [30] reported no technical difficulties. The studies by Kallio et al. [28] and Sood and colleagues [30] did not mention whether or not technical difficulties were encountered. Of the adult studies mentioned [23,32,33] only the study by Hadfield mentioned difficulty in acquiring and maintaining the NAVA signal in 10 of 36 patients (28%). This suggests that technical difficulties are relatively rare and may be dependent upon operator experience and preference.

None of the included studies reported effects on cost of care with NAVA use. To upgrade an existing Maquet/Getinge Servo-i ventilator to NAVA requires additional software, hardware, and a dedicated single-use Edi catheter per patient. Additional training is required, usually provided by the ventilator company. One paper by Hjelmgren et al [36] modelled the potential for NAVA to save costs based on evidence of improvements in the asynchrony index in adult studies of NAVA. From our own estimate, based on the UK reference costs of NAVA software and the hardware cost for one Maquet Servo-i ventilator of GBP 9600 [37], and based on an average 8-year working life of a PICU ventilator, this equates to a daily cost of GBP 3.28. The cost of a NAVA Edi catheter is GBP 145. These costs do not include value-added tax (VAT). The 2017–2018 reference cost of a UK PICU bed day is GBP 2178. We have not estimated costings for the training time of PICU staff. However, based on these estimates, and a reduction of PICU length of stay of around 24 h, as observed in the RCT by Kallio et al. [27], NAVA is likely to be significantly cost effective in terms of PICU care costs. The available evidence is insufficient to make recommendations regarding the routine use of NAVA for the weaning of invasively ventilated children. Further research is required, focussed on relevant clinical outcomes in adequately powered clinical trials, which should include full health economic evaluations.

The evidence included in our systematic review had several limitations. We found only one RCT [27]. The numbers included in the cohort studies [29,30] were relatively small and this may have underestimated any effects of NAVA. The studies included differing PICU populations. The RCT by Kallio et al. [27] involved all-comers with a predominance of post-operative patients. The cohort studies included PARDS patients and children recovering from heart surgery on cardiopulmonary bypass. The effects of NAVA may be different in different PICU subpopulations. Also, outcome measures were not directly comparable as they were expressed in different units between stud-ies. None of the included studies addressed cost.

Our review process had the following limitations. We limited our search to Med-line, CINAHL and EMBASE databases, and the Cochrane Library and TRIP database and did not perform hand searching. We may have missed other studies not included in these databases. Due to the few included studies with only one RCT it was not possi-ble to perform any data synthesis of outcomes.

## 5. Conclusions

This systematic review of NAVA versus conventional ventilation weaning modes in invasively ventilated children revealed weak evidence of efficacy in terms of improved clinical outcomes, especially a shorter duration of ventilation and PICU length of stay, as well reduced sedative use. We found no evidence for an improved cost of care and are unable to make recommendations regarding the routine use of NAVA in the weaning of invasively ventilated children in PICUs. The data summarised in this review may be valuable in planning further adequately powered RCTs with health economic evaluations, in order to address these important unanswered questions.

## Figures and Tables

**Figure 1 jcm-10-03393-f001:**
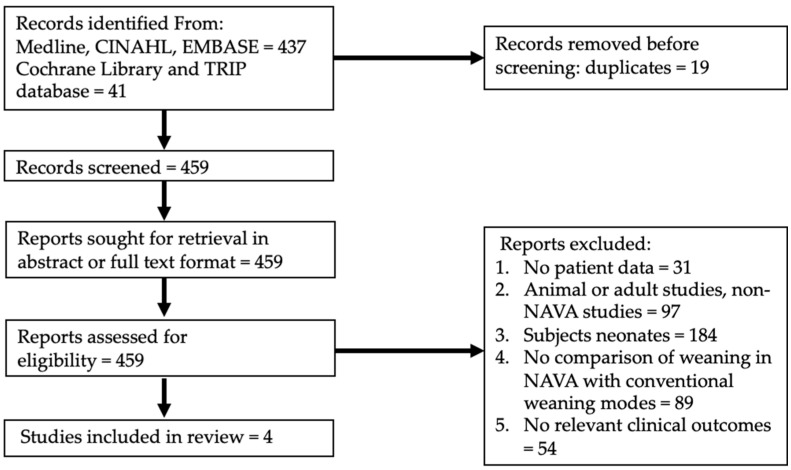
Flow diagram for systematic review of NAVA versus conventional weaning modes in children.

**Table 1 jcm-10-03393-t001:** Included studies of weaning in NAVA versus conventional modes of ventilation in children.

Reference	Study Population	Type of Study	Outcome Measures					
			Duration of Weaning	PICU Length of Stay	Sedative Use	Re-Intubation within 24 h	Mortality	Complications of Ventilation
Kallio 2015 [27]	Invasively ventilated children 0–16 years, *n* = 170	Randomised Controlled Trial	Median 3.3 versus 6.6 h (NS)	Median 49.5 versus 72.8 h (*p* = 0.03, per-protocol analysis)	Mean difference in sedation units −1.43 units (95% CI −2.79 to −0.07; *p* = 0.03)	No difference *n* = 3 (NAVA group) vs. *n* = 4 (control)	No difference *n* = 0 (NAVA group) vs. *n* = 2 (control)	Accidental extubation *n* = 1 vs. *n* = 2
Piastra 2014 [29]	Children ventilated for ARDS, *n* = 30	Cohort	Mean 41 (±17) versus 72.5 (±44) h (*p* = 0.011)	No difference	COMFORT score mean 18.1 (±2.1) vs. 25.3 (±7, *p* = 0.004) for same average dose	Not reported	No deaths during study period, 2 later deaths in each cohort	Not reported
Sood 2019 [30]	Children recovering following cardiac surgery on CPB, *n* = 75	Cohort	Median total ventilation days 9 (IQR 4) vs. 11 (IQR 7.5, *p* = 0.0128)	Median 9 days (IQR 4) versus 13.5 days (IQR 7.5, *p* < 0.0001)	Fentanyl days 9 (IQR 5) vs. 12.5 (IQR 7, *p* < 0.0001) Midazolam days 8 (IQR 4) vs. 12 (IQR 7, *p* < 0.0001)	2.9 % vs. 20% (*p* = 0.032), OR for successful extubation 8.5 (95% CI 1.01 to 71.8)	Not reported	Not reported

Abbreviations: NS = not significant; 95% CI = 95% confidence interval; CPB = cardiopulmonary bypass; IQR = interquartile range.

**Table 2 jcm-10-03393-t002:** Risk-of-bias assessment for Kallio et al., 2015 [27].

Source of Bias	Description	High Risk	Low Risk	Unclear	Comment
Random sequence generation	Randomisation by computerised random-number generator and opaque envelopes		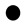		
Allocation concealment				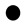	Not described in sufficient detail
Selective reporting				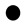	Insufficient information
Blinding(participants and personnel)	Due to the nature of NAVA intervention, it would not have been possible to blind participants and personnel	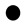			
Blinding(outcome assessment)				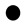	Not described in sufficient detail
Incomplete outcome data	Of randomised patients, 3 patients in the NAVA group and 2 controls did not enter the study. Missing data not specifically reported			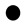	Not described in sufficient detail
Selective reporting	All prespecified outcomes were reported		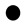		
Other sources of bias				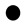	Insufficient information

## Data Availability

The data presented in this study are available on request from the corresponding author. The data are not publicly available due to the data being held on an institutional drive, with restricted access.

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
