# Peer review of "Neurally-Adjusted Ventilatory Assist (NAVA) versus Pneumatically Synchronized Ventilation Modes in Children Admitted to PICU"

_jcm, 2021, doi:10.3390/jcm10153393_

Round 1

Reviewer 1 Report

It is an interesting review, since NAVA is not a commonly available mode of ventilation.  Most of the aspects,that are important from the practical point of view are discussed.

It could be interesting to know if there were any technical difficulties with the use of NAVA mentioned in the cited references. It could be helpful for those who aer thinking about starting to use this method.

Author Response

Dear Reviewer,

Thank you for reviewing our paper and for your feedback.

In response to your question about technical difficulties with NAVA in the included studies, the studies by Kallio (reference 27) et al and Sood and colleagues (reference 30) did not specifically report on this. Piastra and colleagues (reference 29) specifically mentioned that there were no technical difficulties with NAVA.

In the feasibility study of adults at risk of prolonged mechanical ventilation by Hadfield et al (reference 23) in 10 of 36 patients (28%) difficulties were encountered in acquiring and maintaining the Edi signal. Both Kacmarec et al and Liu et al (references 32 and 33) reported no adverse events related to use of NAVA.

In conclusion, it appears that technical difficulties are relatively rare and may be influenced by user experience and preference of weaning mode.

We have added this to the discussion.

We look forward to hearing from you.

Best wishes,

Reinout Mildner

Reviewer 2 Report

Overall, a very well written systematic review of Neurally adjusted ventilatory assist (NAVA) versus pneumatically synchronized ventilation mode in pediatric population. The selection process of studies was thorough and appropriate. Although significant number of studies were excluded and only 4 studies were reviewed, it is appropriate as this is a systematic review and not a proper meta-analysis. 

The authors appropriately provide a narrative review of the findings in the selected studies and conclusion appears appropriate. They highlight drawbacks and limitations of their review which is nicely done.

Only suggestion: Use full form when using abbreviations for the first time-more uniformly. 

Author Response

Dear Reviewer,

Thank you for reviewing our paper and for your feedback.

You have suggested to use full form when using abbreviations for the first time-more uniformly.

We have checked the entire paper and ensured that all abbreviations are preceded by their full form, apart from GBP and COMFORT sedation score. We have also corrected a small number of typing errors. These are all tracked in red in the updated version of the paper.

Please let us know if you have any further questions or comments.

Best wishes,

Reinout Mildner